# Vav Proteins in Development of the Brain: A Potential Relationship to the Pathogenesis of Congenital Zika Syndrome?

**DOI:** 10.3390/v14020386

**Published:** 2022-02-14

**Authors:** Aidan J. Norbury, Lachlan A. Jolly, Luke P. Kris, Jillian M. Carr

**Affiliations:** 1Flinders Health and Medical Research Institute, College of Medicine and Public Health, Microbiology and Infectious Diseases, Flinders University, Bedford Park, Adelaide, SA 5042, Australia; anorbury@flinders.edu.au (A.J.N.); luke.kris@flinders.edu.au (L.P.K.); 2Robinson Research Institute, Adelaide Medical School, The University of Adelaide, Adelaide, SA 5000, Australia; lachlan.jolly@adelaide.edu.au

**Keywords:** zika virus, Vav, Rac1, RhoA, neuronal development, congenital zika syndrome, neuronal progenitor cell, brain development, guanosine nucleotide exchange factor

## Abstract

Zika virus (ZIKV) infection during pregnancy can result in a significant impact on the brain and eye of the developing fetus, termed congenital zika syndrome (CZS). At a morphological level, the main serious presentations of CZS are microcephaly and retinal scarring. At a cellular level, many cell types of the brain may be involved, but primarily neuronal progenitor cells (NPC) and developing neurons. Vav proteins have guanine exchange activity in converting GDP to GTP on proteins such as Rac1, Cdc42 and RhoA to stimulate intracellular signaling pathways. These signaling pathways are known to play important roles in maintaining the polarity and self-renewal of NPC pools by coordinating the formation of adherens junctions with cytoskeletal rearrangements. In developing neurons, these same pathways are adopted to control the formation and growth of neurites and mediate axonal guidance and targeting in the brain and retina. This review describes the role of Vavs in these processes and highlights the points of potential ZIKV interaction, such as (i) the binding and entry of ZIKV in cells via TAM receptors, which may activate Vav/Rac/RhoA signaling; (ii) the functional convergence of ZIKV NS2A with Vav in modulating adherens junctions; (iii) ZIKV NS4A/4B protein effects on PI3K/AKT in a regulatory loop via PPI3 to influence Vav/Rac1 signaling in neurite outgrowth; and (iv) the induction of *SOCS*1 and *USP9X* following ZIKV infection to regulate Vav protein degradation or activation, respectively, and impact Vav/Rac/RhoA signaling in NPC and neurons. Experiments to define these interactions will further our understanding of the molecular basis of CZS and potentially other developmental disorders stemming from in utero infections. Additionally, Vav/Rac/RhoA signaling pathways may present tractable targets for therapeutic intervention or molecular rationale for disease severity in CZS.

## 1. Introduction

Zika virus (ZIKV) is a positive-sense single-stranded RNA virus of the flavivirus genus, which includes other viruses such as dengue virus, West Nile virus, Japanese encephalitis virus and yellow fever virus [1,2]. Although ZIKV was first identified in 1947, the virus was largely restricted to Africa and had little clinical impact [3,4]. Since 2007 however, ZIKV outbreaks have been documented in French Polynesia, Brazil, parts of Southeast Asia such as the Philippines and Yap Island, and even parts of the USA such as Florida [1,3,5,6]. Although predominantly transmitted by *Aedes* spp. mosquitoes, ZIKV can also be transmitted between humans through blood, sexual contact and vertically [2,7,8].

Approximately 80% of adult ZIKV infections are asymptomatic, but in those adults who are symptomatic, the clinical manifestations include fever, rash, myalgias/arthralgias, malaise and headaches, lasting for up to 7 days [4,8,9]. ZIKV infection is also associated with Guillain Barre syndrome (GBS) in adults—a post-infectious inflammatory neurological condition that can cause paralysis and life-threatening respiratory compromise [10]. The most concerning impact of ZIKV, however, is an infection during pregnancy and exposure in utero.

Maternal ZIKV infection in any of the three trimesters of pregnancy can result in vertical transmission of ZIKV to the developing fetus where adverse outcomes can ensue [7]. This is of particular importance to the developing brain and eye [7,11,12,13]. Congenital zika syndrome (CZS) is the term used to describe the cluster of structural and functional neurological manifestations of fetal ZIKV infection [9,14]. CZS principally involves damage to the fetal brain, reduced intracranial volume and cranial abnormalities, which can ultimately lead to manifestations such as global neurodevelopmental delay, epilepsy, deafness and visual impairment [9,14]. Prominent cranial abnormalities include reduced intracranial volume leading to microcephaly, overlapping cranial sutures, depression of frontal and parietal bones and prominence of the occiput [9,14]. Brain imaging and post-mortem studies of CZS brains have revealed neuroanatomical abnormalities such as cortical thinning, reduced prominence of gyri, ventriculomegaly, cortical/subcortical calcifications, hypoplasia of the corpus callosum and cerebellum and other abnormalities of the white matter [9,14,15]. At a cellular level, neuronal cell death, hyperplasia of microglia, myelination delay and gliosis/glial scarring have been demonstrated [16]. Defective eye development is also a hallmark of CZS [17]. Ocular pathologies such as macular scarring and retinal mottling, microphthalmia, intraocular calcification, optic nerve abnormalities (cupping, pallor, hypoplasia), chorioretinal atrophy, subretinal hemorrhage and vascular abnormalities have been reported [17,18]. Beyond these initial manifestations present at birth and early postnatal life, more subtle neurodevelopmental manifestations may arise with age. One prospective study found that around a quarter of babies potentially exposed to ZIKV in utero and who did not exhibit symptoms or signs at birth, eventually progressed to developing visual, auditory or developmental abnormalities [19]. This suggests that CZS involves a spectrum of developmental defects in the brain and eye, and even in the absence of gross morphological changes, there are significant alterations that have a functional impact.

The underlying pathogenic mechanisms of CZS at a cellular level remain incompletely understood. There is a suggestion that the pathogenesis of CZS may in-part involve inflammatory changes and dysregulation of interferons in the developing brain [20,21]. Microglia and astrocytes play key roles in central nervous system (CNS) development and homeostasis and mediate CNS inflammatory responses [22]. These cells are also susceptible to ZIKV infection, suggesting that their dysfunction may contribute to CZS [23]. Transcriptional profiling in a mouse model of ZIKV microcephaly has identified many genes dysregulated that are associated with neurogenesis, cell death and microglial activation [24]. It is, however, the ability of ZIKV to target and infect the developing brains’ primordial cells, called neural progenitor cells (NPCs), and the neurons they generate during fetal stages of CNS development, which is considered the key driver of CZS pathogenesis [23,25,26,27]. ZIKV can disrupt multiple NPC functions including cell division, survival and differentiation, and impair the function of their neuronal derivatives, including neuronal migration, outgrowth and viability [13,25,26]. This has been discussed in the broader comparative context of the effect of neurotropic viruses such as herpesviruses, human immunodeficiency virus (HIV) and ZIKV on neural stem cells [28]. Whilst the impact of ZIKV infection on the function of NPC and neurons is established, the molecular mechanisms driving dysfunction remain to be elucidated. It is intriguing that the cellular pathologies of CZS are processes known to be regulated by Vav proteins.

Vavs are a family of proteins that function as guanine exchange factors (GEF) that activate Rho family GTPases, such as RhoA, Rac1 and Cdc42 [29,30,31,32]. The Rho family GTPases and Vav regulatory proteins have well-characterized roles in controlling the function of NPCs and developing neurons [33,34,35,36,37,38]. This review focuses on known roles for Vav proteins in neurodevelopment and several points of potential molecular interaction between ZIKV and Vav signaling pathways that we propose may influence the pathogenesis of CZS.

## 2. Vav Proteins and Vav Signaling Pathways

There are three members of the Vav family of proteins in mammals—Vav1, Vav2 and Vav3—which exhibit similar but not identical functions, as reviewed elsewhere [29,32,39,40]. In brief, Vav1 is expressed almost exclusively in cells of hematopoietic origin, whereas Vav2 and Vav3 are more ubiquitous [40,41,42]. There has been considerable focus on Vavs, particularly Vav1 in immune cell functions, with key roles in T and B-lymphocyte receptor signaling and macrophage driven inflammatory responses [29,32,42]. Vav2 is implicated in endothelial cell function, vasodilation and blood pressure regulation [43,44]. Vavs have also been implicated in various aspects of cancer biology, such as tumorigenesis, epithelial-mesenchymal transitions, metastases and chemosensitivity [44,45,46,47,48]. Relevant to CZS, Vav2 and Vav3 are expressed in the developing embryonic brain and retina and play important roles in NPC function and differentiation, and neuron cell growth, maturation and synapse formation, as discussed further below [36,49]. *VAV*2 in particular displays a strong developmental transcript expression pattern across all regions of the human brain, with high levels of expression in utero, which is dramatically downregulated postnatally, suggesting a fundamental role in early brain development (Figure 1). This is supported by studies demonstrating decreasing *Vav*2 mRNA with time in the hippocampus throughout in utero development (36), the presence of *Vav*2 protein in early postnatal retinal ganglion cells, embryonic cortical and hippocampal neurons and a decrease in Vav2 protein with time in brain lysates throughout embryonic and early postnatal development in the mouse (51).

At a molecular level, Vavs function intracellularly in cellular signaling pathways, downstream of receptors with either intrinsic or associated tyrosine kinase activity [32,40] (Figure 2). These receptors include, but are not limited to, antigen receptors, chemokine receptors and Fc receptors. Importantly, Vav also signal downstream of TAM receptors (Tyr, Axl and MerTK), which can be targets for ZIKV entry [50] and the tropomysin receptor (TrkB), the receptor for brain-derived neurotropic factor (BDNF), which has a key role in neurogenesis and synaptic development [36,51,52,53,54,55,56]. Receptor stimulation results in tyrosine phosphorylation of Vav either directly or indirectly via non-receptor-associated kinases such as Src and Syk (Figure 2) [57]. The phosphorylation of Vavs induces a conformational change which permits Vav GEF catalytic activity [57]. Vav can also be activated by non-tyrosine phosphorylation-dependent mechanisms such as by binding to phosphatidylinositol 3,4,5-triphosphate (PIP3) [58,59] (Figure 2). The activation of Vavs in turn enables them to activate Rho family proteins including Rac1, RhoA and Cdc42, by mediating the exchange of bound GDP for GTP [39,40] (Figure 2). The GTP-bound Rho proteins can subsequently interact with other effector proteins as part of cellular signaling pathways [30], with molecular targets including kinases, scaffold proteins, cytoskeletal regulatory proteins and adaptor proteins [60,61]. While there are also non-catalytic functions of Vav proteins, most of the many diverse physiological roles of Vavs are driven by GEF activation of Rho family proteins and their subsequent effects on the cytoskeleton [29,32]. The modulation of the cytoskeleton following Vav/Rho GTPase signaling is integral for numerous cell functions such as vesicular transport, phagocytosis, cellular growth, adhesion, migration and chemotaxis [62,63,64,65,66,67]. Importantly, Vav-Rho GTPase-mediated modulation of the cytoskeleton is also essential for various aspects of NPC and neuronal cell function, and neurodevelopment in general, as highlighted below and summarized in Figure 2 [49].

## 3. Vav Proteins and Neurodevelopment

The development of the brain is a complicated and tightly regulated process [68,69]. The mature brain consists of a vast array of different types of neuronal cells and supporting glia and oligodendrocytes [70]. These cell types arise in an orchestrated manner during brain development [70]. The entire CNS arises from a primordial NPC population known as neuroepithelial NPCs, which are organized in a tube structure called the neural tube [71]. As these cells expand, they become patterned both spatially and temporally into distinct NPCs with a restricted function and fate [72]. The conversion of neuroepithelial cells into a second type of NPC known as a radial glial cell (RGC) accompanies the onset of neurogenesis—the differentiation of RGCs into post-mitotic neurons [71,73]. Like neuroepithelial NPCs, RGCs display overt apical-basal polarity which is essential for function [74]. RGC apical membranes demarcated by cell-adhesion complexes define the ventricular surface of the neural tube [74]. The RGC cell bodies occupy the proliferative ventricular zones (VZ) of the developing brain, with their basolateral processes acting as a structural scaffold that guides the migration of differentiated cells away from the VZ to the correct location in the developing brain [75,76,77,78]. The RGCs must balance self-renewal and differentiation to achieve the correct size and cellular diversity of the brain, and this is driven by the signaling environment within the VZ that instructs intrinsic genetic programs [79,80]. As such RGC cell adhesion, polarity and self-renewal are interdependent and the normal coordinated loss in RGC during development ensues due to their exit from the VZ and execution of a genetic program of differentiation. During development, RGCs initially produce neurons, and after a wave of neurogenesis, RGCs undergo subsequent waves of gliogenesis and finally oligodendrogenesis to form other cell types in the brain [73,81]. Following their birth, these new cells undergo extensive migration and maturation processes that are driven by signaling cues from the brain environment [68]. For example, maturation of neurons involves growth and arborization, or branching of dendrites and axons, synapse formation and generation of functional circuitry. At the conclusion of development, the vast majority of NPCs are exhausted, and the spatial localization and timing of the generation of the differentiated neurons, glia and oligodendrocytes drives their final identity, which underpins the diversity, neuronal circuitry and overall brain architecture [68,69,73].

### 3.1. Vav and Rho GTPase Signaling in Ventricular Zone NPC Localization and Formation of Adherens Junctions

As mentioned above, NPCs maintained within the VZ are self-renewing and there is extensive literature linking the Rho signaling pathway to the polarity, adhesion and VZ localisation of NPCs and therefore their self-renewal [34,35,82]. Conditional deletion of the Rho GTPase, *Cdc*42, leads to a progressive loss of adherens junctions and apico-basal polarity and is associated with differentiation and exit of VZ NPCs [34]. The outcome of this premature differentiation is a reduction in the size of the brain [34]. Likewise, *Rac*1 knockout results in disrupted VZ architecture leading to premature NPC differentiation associated with increased cell cycle exit and reduced NPCs proliferation and self-renewal [83]. Conditional genetic deletion of *Rac*1 from the VZ of the telencephalon recapitulated this phenotype, and further demonstrated increased apoptosis [35]. This premature differentiation culminated in reduced size of the cortex and striatum, resembling microcephaly [35]. Similarly, conditional *Rho*A deletion in the spinal cord of embryonic mice resulted in reduced NPC pools via reduced proliferation and increased cell-cycle exit [82]. The presence of abnormal infiltration of stem cells into the ventricle of the spinal cord, is indicative of a loss of NPC adhesion and polarity characteristics [82]. Conditional *Rho*A knockout in the midbrain and forebrain of mice also resulted in a disruption to NPC adhesion and polarity of VZ NPCs [37]. Intriguingly, in these mice ectopic neural tube-like structures called neural rosettes arose at basal locations which drove hyperproliferation of NPCs [37]. Both these studies reveal a disruption of NPC adherens junctions, driving divergent dysplastic changes within their respective tissues in the absence of RhoA [37,82]. Collectively, these data suggest that loss of Rho function leads to loss of NPC adhesion, polarity and premature differentiation at the expense of NPC expansion, therefore influencing the size and architecture of the developing brain tissues. This is relevant to CZS, as it is generally appreciated that microcephaly—a defining feature of CZS—is caused by overlapping mechanisms which reduce the capacity of NPC proliferation or drive premature differentiation with associated cellular demise [24,84,85]. Although activation of Rho family GTPases can be achieved by numerous GEFs, Vav GEF may be involved as a conduit between the extrinsic signaling environment in the VZ and the control of NPC function through Rho GTPase activation (Figure 2).

Evidence for this mechanism comes from studies in the developing eye [86]. Live cell imaging of Vav-deficient drosophila eyes demonstrated that stable adherens junctions did not form between primordial cells in the developing ommatidia, resulting in disorganized architecture, ectopic specification of photoreceptors and overall excess of photoreceptors in the eye [86,87]. In mice, *Vav*3 knockout exhibited an accelerated differentiation of retinal NPCs into cone and retinal ganglion cells, and more cells exhibiting markers of late retinal progenitor cells [88]. This supports functions of Vav in maintaining a progenitor cell pool and preventing premature differentiation in the eye and raises the possibility of a common Vav-Rho GTPase-mediated effect in NPC in the brain.

### 3.2. Vav and Rho GTPase Signaling in Neuronal Maturation

Following their production during neurogenesis, neurons must mature to form functional circuitry within the brain. A major event in maturation is neuritogenesis, the growth and arborization of neurites including dendrites and axons that enable them to project from the neuronal cell body and connect with other neurons throughout the brain. A key structure driving this process is the growth cone, located at the leading edge of outgrowing neurites [89]. The development and function of growth cones is largely dictated by modulation of internal cytoskeletal structures including filopodia (F-actin bundles), lamellipodia (actin mesh-like structures) and microtubules [89]. A balance of attractive and repulsive molecular signals at the growth cone is required to perform directional outgrowth and enables neurites to navigate throughout the brain to their desired targets [90]. Attractive neuritogenic signals induce growth cone membrane protrusion towards a target, through growth cone actin polymerizations whereas repulsive molecular signals induce growth cone collapse, involving stabilization and/or depolymerization of F-actin filaments [89]. These processes are controlled by combinations of extracellular signaling molecules, such as fibroblast growth factors, SLIT, Ephrin, NGF and BDNF [89,91,92,93,94,95]. A key concept in growth cone regulation is that stimuli which activate Rac1 and Cdc42 drive actin polymerization in the growth cone leading edge to promote neurite outgrowth, whereas stimuli that activate RhoA, stabilize F-actin and the cytoskeleton and either stall neurite outgrowth or facilitate ‘chemo-repulsive turning’ of outgrowing axons [94,96,97,98] (Figure 1). The Vavs, in particular Vav2 and -3, have been implicated in mediating the effects of several signaling pathways that control neurite outgrowth [33,51,58,97]. Experiments using organotypic hippocampal sections and cultured cells demonstrate that BDNF stimulation of TrkB receptors activates Rac1 via Vav2 or -3 to promote neurite outgrowth [36]. Likewise, activation of the CD47 receptor by integrin binding results in Src-dependent phospho-activation of Vav2 in primary hippocampal neurons [38]. The resulting downstream activation of Rac1/Cdc42 promotes actin reorganization and underpins both neurite growth and formation of dendritic spines—a precursor for synapse formation [38]. Control of neurite growth by NGF also involves the Vav-Rho axis, where stimulation of neural-based PC12 cells with NGF resulted in downstream activation of PI3K, accumulation of PIP3, which recruits and activates Vav2 and Vav3 at the plasma membrane [58,59] (Figure 1). Vav2/3 subsequently activates Cdc42/Rac1 to promote neurite outgrowth [58]. In this model, PIP3 accumulation is dependent on Vav2, illustrating a regulatory loop that reinforces neurite outgrowth [58]. The role for Vav2/Rac1 in this process is also evident from studies of *Xenopus* spinal neurons, where Vav2 overexpression inhibits RhoA signaling and promotes neurite growth [94].

By mediating neurite growth cone growth or collapse, Vav also regulate signals that appropriately guide and target axons as they navigate through the brain. The cell–cell signaling between Eph and Ephrin receptors is a well-established axon guidance mechanism and involves activation of Vav [51,99,100]. The Ephrin/Eph interaction between an axon and neighboring cells is initially adhesive [51]. This adhesive interaction then recruits Vav2 to the intracellular domain of Eph where Vav activates Rac1 [51]. In this context, activated Rac1 drives endocytosis of the Eph/Ephrin complex, growth cone collapse and subsequent repulsion of the axon such that it continues to navigate to find its correct target [51]. Furthermore, *Vav*2 and *Vav*3 knockout mice exhibit fewer ipsilateral retinogeniculate axonal projections and demonstrate skewing of axon positioning in the thalamus [51]. Several other studies have also revealed axonal guidance defects in the absence of Vav. A role for Vav3 specifically in axon guidance of GABAergic neurons in the brainstem has been demonstrated [101]. Immunohistochemistry on *Vav*3^−/−^ mouse brains revealed reduced expression of GABAergic axon terminal markers in the ventrolateral medulla (VLM), while axonal tracing revealed a failure of caudal VLM GABAergic axonal projections to properly innervate their targets in rostral aspects of the VLM [101]. Further, guidance errors in the outgrowth of Vav2 knockdown commissural interneuron axons were observed, with the angle of midline crossing in these neurons being more random and less organized than neurons of wild-type Xenopus [94]. In drosophila, where there is only one Vav isoform, Vav is necessary during development of the eye for mediating the guidance of ipsilateral photoreceptor axons from the eyes to the optic lobes, with Vav mutants exhibiting abnormal patterns of axon targeting in the lamina and disruptions to the retinotopic map in the medulla [87,102].

Collectively, these data implicate Vav signaling pathways in a range of processes for neuronal cell maturation including neurite outgrowth and axon targeting in a variety of developing CNS tissues including the eye. The major site of Vav activity is in the growth cone, and the mechanisms of Vav action typically converges to the regulation of Rho, Rac1 and Cdc42. These same neuronal maturation processes are disturbed in CZS, and this places Vavs as candidate mediators of CZS pathogenesis.

## 4. Proposed Links between Vavs and ZIKV-Induced Neuropathology

### 4.1. Stimulation of Vav-Mediated Signaling Directly by ZIKV Binding and Entry?

Vav may be linked to the pathogenesis of CZS through the process of viral entry into cells of the CNS. The TAM receptors although not obligatory for infection, can mediate ZIKV entry into cells including NPC, microglia and astrocytes [50,103,104,105]. Since Vav can mediate signaling downstream of these receptors, TAM-stimulated Vav-signaling may be affected by ZIKV interaction with these receptors [50]. For example, the extracellular ligand Gas6 which is present in the developing brain microenvironment, forms a complex with ZIKV particles and assists in mediating the interaction of ZIKV with TAM receptors, such as Axl [50]. The interaction of Gas6 with Axl/MerTK stimulates phosphorylation of Vav and subsequent Rac1 activation leading to stimulation of phagocytosis in microglial cells [106]. This interaction of TAMs with Vav is also supported in monocytes where Vav1 physically interacts with and is phosphorylated following activation of MerTK [107]. There is precedence for viral triggering of Vav signaling, with signaling in HIV-infected T-cells via the T-cell receptor activating Rac1/Cdc42 via a direct interaction of the HIV nef protein with Vav [108]. Further, the important roles of Rho GTPases in viral entry and exit, primarily through changes to the cytoskeleton, has been reviewed [109]. Specifically, for flaviviruses, Japanese encephalitis virus (JEV) infection in neuronal cells rapidly activates RhoA and blocking RhoA reduces JEV entry and infection [110]. Several studies have described effects of Rac1 and Cdc42 on DENV entry and virus production that is linked to the PI3K/AKT pathway and actin cytoskeletal changes [111,112,113], although as yet, there are no investigations to assess the role of Vav in activation of these GTPases during the flavivirus entry process. Similar roles for Rho GTPases in ZIKV receptor binding and entry are likely, potentially following ZIKV–TAM interaction, with Vav acting as a signaling intermediate in this response (Figure 3A).

### 4.2. ZIKV-NS2A Disruption of Vav-Regulated Adherens Junctions in Developing Neural Cells/NPC?

ZIKV-NS2A expression in developing mouse cortices and ZIKV infection of developing brains reduces proliferation and induces premature differentiation of RGCs, as well as aberrant neuronal positioning and cortical arrangement [114]. Specifically, ZIKV-NS2A expression reduces ZO1, beta-catenin, SMDA7 and NUMBL protein levels, all of which are key adherens junction components [114]. ZIKV-NS2A coimmunoprecipitates with these adherens junction proteins, as well as N-cadherin [114]. RhoA and Vav have important roles in regulating NPC and axonal interactions through effects on adherens junctions [37,86], with Vav itself hypothesized to be part of the adherens junction complex, serving as a link to the actin cytoskeleton [86]. Hence, although there is no data to support direct interactions of Vav with ZIKV proteins, there is a functional convergence of Vav and ZIKV NS2A to modulate adherens junctions in an opposing manner (Figure 3B).

### 4.3. ZIKV-NS4A/B Disruption of PI3K/AKT Signaling Pathways?

Other ZIKV proteins have been implicated in disrupting neural stem cell development and autophagy through the PI3K/AKT pathway [115]. Specifically, the expression of NS4A and/or NS4B in human fetal neural stem cells inhibits the phosphorylation and activation of AKT and ultimately hinders the differentiation of fetal neural stem cells into mature neurons [115]. The PI3K/AKT pathway, via PIP3, drives Vav/Rac1/Cdc42 signaling in a regulatory feedback loop (Figure 1 and Figure 2) [38,46,58,82]. This raises the proposal that ZIKV NS4A/4B may inhibit PI3K/AKT resulting in reduced PIP3-mediated activation of Vav and a subsequent reduction in Rac1/Cdc42 signaling and neurite outgrowth (Figure 3C).

### 4.4. SOCS1-Mediated Degradation or USP9X-Mediated Activation of Vav?

A normal and important response to viral infection is the induction of interferon (IFN) and inflammatory pathways, with the JAK/STAT signaling pathway key to many of these immune/inflammatory responses. This pathway can be influenced by Rho GTPases, such as Rac1 in response to IFN-γ [116] and the innate TLR3- and IFN-driven responses, resulting in impaired neurogenesis [117]. Additionally, these pathways are downregulated in a temporal manner by suppressors of cytokine signaling (SOCS) 1 and 3 to resolve these early innate cytokine/antiviral responses [118,119]. Following ZIKV infection of NPCs, there is increased expression of *SOCS*1 and 3 and a reduction of type I and III IFN responses [119]. SOCS1 has been shown to directly interact with Vav via its SH2 domain, in a phosphorylation-independent manner, and this subsequently targets Vav for ubiquitination and degradation [120]. Vav also interacts and is activated by the adaptor molecule ZAP70 [39,121], which is in turn activated through deubiquination by USP9X [122]. USP9X has a key role in maintaining NPC self-renewal properties and is linked to neurodevelopmental disorders [123,124]. The mRNA for *USP9X* is increased by ZIKV infection [125] and thus it could be reasonably hypothesized that following ZIKV infection, the balance between SOCS1, driving Vav ubiquitination, degradation and loss of Vav function, with USP9X, driving ZAP70 deubiquitination and activation of Vav, could regulate Vav/Rho GTPase activation and signaling in NPC and neurons (Figure 3D).

## 5. Conclusions and Future Perspectives

Vav proteins have been shown to be key players in multiple cellular and physiological processes, including neurodevelopment. In particular, Vav/Rho GTPase signaling pathways influence NPC function including their cell adhesion, polarity, proliferation, differentiation fate and survival. Additionally, Vav/Rho GTPase signaling affects neuronal cell functions including neurite growth cone regulation, axon projection and targeting. This presents a circumstantial association of Vav/Rho GTPase signaling pathways with the pathogenesis of CZS—where these aspects of neurodevelopment are also disrupted. Potentially, ZIKV may influence Vav/Rho GTPase signaling by stimulating TAM-Vav-dependent signaling pathways following ZIKV entry; ZIKV alterations in Vav-regulated adherens junction formation; ZIKV inhibition of Vav through inhibition of PI3K/AKT pathways and reduced production of PIP3; or via SOCS1-mediated ubiquitination and degradation, or USP9X-mediated activation of Vav proteins.

Further investigation of these interactions is warranted to define the molecular pathways driving the pathogenesis of CZS and potential therapeutic targets to prevent NPC and neuronal cell dysfunction. These outcomes may also have an impact on understating of infection-driven neurodevelopmental pathologies that impact the NPC pool [28]. Additionally, commonly available and clinically utilized drugs, such as azathioprine and ibuprofen can inhibit Rac1 activation by Vav and Vav phosphorylation, respectively [126,127]. Given the potential roles of Vav/Rho GTPase signaling described in this review, such agents may be contraindicated during ZIKV infection in pregnancy.

Additionally, there are known polymorphisms in *VAV*s that can influence infectious diseases, such as susceptibility to *Candida albicans* [128]. Given the considerable spectrum of pathology of CZS and the little understanding of what predicts such variability, further research of Vav and *VAV* polymorphisms in relation to CZS could lead to useful clinical management strategies or prognostic information.

## Figures and Tables

**Figure 1 viruses-14-00386-f001:**
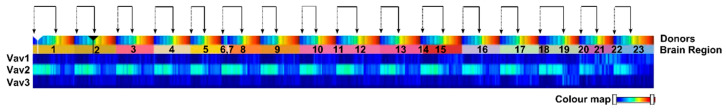
Transcriptional profile of *VAV* in the developing human brain. *VAV* expression was analyzed using the developmental transcriptome provided by the Allen Institute for Brain Science, Allen Brain Atlas, Brain Span, Atlas of the developing human brain https://www.brainspan.org (accessed on 4 January 2022). *VAV*1, -2 and -3 mRNA intensity is assigned in accordance with the color map derived from data for donor tissue from 8–37 pcw, as indicated by the arrows and represented by the blue-green donors, and 4mth–40 yrs represented by the yellow-dark red donor bars. Data is grouped by brain region with 1 = dorsolateral prefrontal cortex; 2 = ventrolateral prefrontal cortex; 3 = anterior cingulate cortex; 4 = orbital cortex; 5 = primary motor cortex; 6/7 = primary motor-sensory cortex; 8 = primary somatosensory cortex; 9 = posteroventral parietal cortex; 10 = primary auditory cortex; 11 = temporal neocortex; 12 = posterior superior temporal cortex; 13 = inferolateral temporal cortex; 14 = occipital neocortex; 15 = primary visual cortex; 16 = hippocampus; 17 = amygdaloid complex; 18 = medial ganglionic eminence; 19 = striatum; 20 = dorsal thalamus; 21 = medial dorsal nucleus of the thalamus; 22 = upper rhombic lip; 23 = cerebellar cortex.

**Figure 2 viruses-14-00386-f002:**
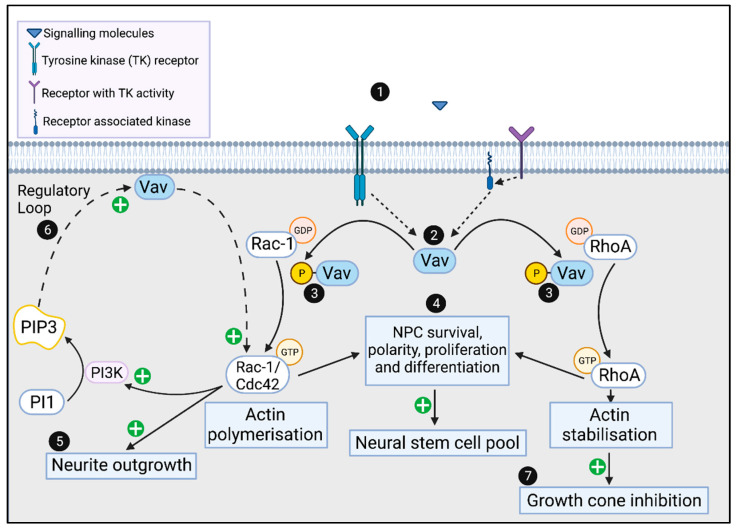
Vav and Rho family GTPase signaling in NPC and neurons. (1) signaling molecules such as NGF, BDNF and Ephrins bind to tyrosine kinase receptors or receptors with associated tyrosine kinases, such as Src/Syk, which can (2) stimulate phosphorylation of Vav. Phosphorylated Vav acts as a guanosine exchange factor to (3) activate Rho family GTPases (Rac1, Cdc42 and RhoA). In neurons, Rac1, Cdc42 and RhoA are involved in maintaining NPC polarity, regulating survival, proliferation and differentiation, which is important for (4) maintenance of the neural stem cell pool. GTP-Rac1/Cdc42 can mediate neurite outgrowth via actin polymerization within the growth cone (5) and activate PI3K, which generates PIP3 that in a regulatory loop, activates Vav to further activate Cdc42/Rac1 (6). GTP-RhoA inhibits the growth cone/neurite outgrowth through actin stabilization (7). Created with BioRender.com (https://biorender.com (accessed on 7 January 2022).

**Figure 3 viruses-14-00386-f003:**
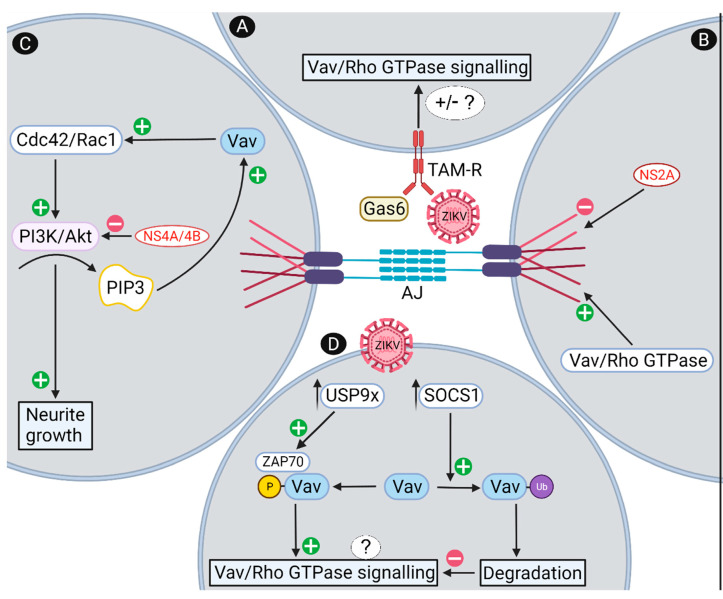
Proposed mechanisms by which ZIKV may affect Vav and Rho GTPase signaling. (**A**) ZIKV can bind and enter cells via TAM receptors, including through an interaction with the TAM ligand Gas6, which can signal via Vav and Rho GTPases with unknown outcomes for this signaling pathway; (**B**) ZIKV NS2A binds to and disrupts adherens junctions, a structure that is regulated by Vav and Rho GTPases, with Vav itself a potential adherens junction component; (**C**) ZIKV NS4A/B inhibits the PI3K/Akt pathway, which is part of a positive feedback loop via PIP3 activation of Vav to drive neurite outgrowth; (**D**) ZIKV induces *SOCS*1 and *USP9X* expression which can affect Vav ubiquitination (Ub) and proteosomal degradation, or ZAP70-induced Vav phosphorylation, which would be predicted to influence Vav activation of Rho GTPases. Predicted outcomes are reduced adherens junctions’ formation and alterations in the actin cytoskeleton that affect the polarization and NPC environment leading to a reduced NPC pool. In neurons, reduced Vav/RhoGTPase signaling is predicted to lead to reduced neurite outgrowth or growth cone collapse, affecting the dynamic changes that influence development of the complex brain architecture. AJ = adherens junctions. Created with BioRender.com https://biorender.com (accessed on 7 January 2022).

## Data Availability

No primary data reported in this publication.

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
