# Peer review of "Vav Proteins in Development of the Brain: A Potential Relationship to the Pathogenesis of Congenital Zika Syndrome?"

_viruses, 2022, doi:10.3390/v14020386_

Round 1
Reviewer 1 Report
In this manuscript, "Vav proteins in Development of the Brain: A potential relationship to the pathogenesis of Congenital Zika Syndrome?", the authors reviewed the recent scientific advances in the field of Congenital Zika Syndrome, specifically the potential link between the Vav proteins and how Zika virus impacts brain development. It contains an interesting hypothesis to be resolved with experimental data. Overall, the review is well-written, and the theme is relevant in the field of developmental neurobiology. However, some minor suggestions could improve the manuscript.
1) Page 3, authors have written: "Relevant to CZS, Vav2 and Vav 3 are expressed in the developing embryonic brain...". It is not clear where and when during brain and retina development these proteins are expressed. This information is crucial for functional interpretation and the possible implications with CZS. A suggestion would be to include the in situ hybridization of the Vav mRNAs or a schematic representation of what is known about Vav protein expression in brain and retina development and co-localization with relevant cell populations (neural progenitors/neurons/glia).
2) Figure 2 could be better represented. It is essential to add some question marks in the schematic scenario (A,B,C and D) where it is still unknown whether Zika infection impacts Vav signalling. Also, it is not clear in scenarios the outcome of Zika virus infection affecting the Vav proteins. For instance, would it decrease neurite growth? The predicted outcomes should be included in the schematic representations. The authors could choose to produce a third figure indicating how loss of function of Vav could impact CNS development (neurogenesis, neuronal polarization, differentiation, axon growth etc).
3) Figure 2: Authors wrote: "(D) ZIKV induces SOCS1 and USP9X.." please include "expression" or overexpression to clarify the sentence.
Author Response
Reviewer 1... Overall, the review is well-written, and the theme is relevant in the field of developmental neurobiology. However, some minor suggestions could improve the manuscript.
1) Page 3, authors have written: "Relevant to CZS, Vav2 and Vav 3 are expressed in the developing embryonic brain...". It is not clear where and when during brain and retina development these proteins are expressed. This information is crucial for functional interpretation and the possible implications with CZS. A suggestion would be to include the in situ hybridization of the Vav mRNAs or a schematic representation of what is known about Vav protein expression in brain and retina development and co-localization with relevant cell populations (neural progenitors/neurons/glia).
Response reviewer 1.1: A new figure has been generated (Figure 1) which is derived from the Allen brain atlas, and summarises the changes in Vav1, -2 and -3 RNA throughout in utero, post-natal and adult development in different regions of the brain. This figure has been interpreted in the text (lines 113-116) and its genesis described in the figure legend (lines 123-135). Additionally, the amended manuscript expands on existing citations (lines 116-121) that have described changes in Vav-2 mRNA and proteins in the brain, with the overall goal of this new data to further support out hypothesis that Vavs, in particular Vav2, are present in the brain and retina and have a developmental function.
2) Figure 2 could be better represented. It is essential to add some question marks in the schematic scenario (A,B,C and D) where it is still unknown whether Zika infection impacts Vav signalling. Also, it is not clear in scenarios the outcome of Zika virus infection affecting the Vav proteins. For instance, would it decrease neurite growth? The predicted outcomes should be included in the schematic representations. The authors could choose to produce a third figure indicating how loss of function of Vav could impact CNS development (neurogenesis, neuronal polarization, differentiation, axon growth etc).
Response reviewer 1.2: Figure 2 (Figure 3 in the amended manuscript) has been modified to clarify unknown scenarios and the figure legends amended to describe predicted outcomes (lines 337; lines 344-349)
3) Figure 2: Authors wrote: "(D) ZIKV induces SOCS1 and USP9X.." please include "expression" or overexpression to clarify the sentence.
Response reviewer 1.3: This has been amended (line 342)
Reviewer 2 Report
In this review, the Authors reviewed the role of Vav proteins in the pathogenesis of Congenital Zika virus Syndrome (CZS). These proteins family have a guanine exchange activity in converting GDP to GTP on proteins such as Rac1, Cdc42 and RhoA to transmit intracellular signaling pathways. They are involved in the developing neurons to control the formation and growth of neurites and mediate axonal guidance and targeting in the brain and retina. The Authors highlighted the possible cellular Vavs proteins-related pathways involved in the Zika virus (ZIKV) infection during vertical transmission, and they identified four potential ZIKV interaction: (1.) binding and entry of ZIKV in cells via TAM receptors, which may activate Vav/Rac/RhoA signaling; (2.) functional interaction of ZIKV NS2A with Vav in modulating adherents’ junctions; (3.) ZIKV NS4A/4B protein effects on PI3K/AKT in a regulatory loop via PPI3 to influence Vav/Rac1 signaling in neurite outgrowth; and (4.) the induction of SOCS-1 and USP9X following ZIKV infection to regulate Vav protein degradation or activation, respectively, and impact Vav/Rac/RhoA signaling in neurons.
Overall, the review is well written and structured: the Vav-related pathways involved in ZIKV infection has been exhaustively discussed and the most relevant references are reported. All these features denote an extensive expertise of the Authors in the field of Flavivirus. However, I would point the attention on few minor revisions to massively improve the already good quality of this review:
- I would incorporate the paragraph The basic principles of neurodevelopment in the next paragraph (Vas and Rho GTPase signaling in ventricular zone NPC localization and formation of adherents’ junctions). This would point the attention of the reader on the main subject of the review.
- The figures 1 and 2 are well made but, if a specific software or app has been used to make them, I would mention it in the captions of the figures.
Author Response
Reviewer 2….Overall, the review is well written and structured: ….However, I would point the attention on few minor revisions to massively improve the already good quality of this review:
- I would incorporate the paragraph The basic principles of neurodevelopmentin the next paragraph (Vas and Rho GTPase signaling in ventricular zone NPC localization and formation of adherents’ junctions). This would point the attention of the reader on the main subject of the review.
Response reviewer 2.1: this has been amended in the revised manuscript (lines 173, 204, 244)
- The figures 1 and 2 are well made but, if a specific software or app has been used to make them, I would mention it in the captions of the figures.
Response reviewer 2.2: this has been clarified in Figure legends 2 and 3 in the revised manuscript with the addition of ‘Created with BioRender.com’. The licence agreement from BioRender is accompanying.
